# OpenReview forum: "Train Short, Test Long: Attention with Linear Biases Enables Input Length Extrapolation"
_ICLR.cc/2022/Conference — ICLR 2022 Poster_

### Official Review · Reviewer_xt6W · 2021-10-27

**Correctness:** 4
**Technical Novelty And Significance:** 3
**Empirical Novelty And Significance:** 3
**Recommendation:** 8
**Confidence:** 4

**Main Review:**

Pros:
- Injecting temporal bias to attention is a neat idea for the language model extrapolation problems.
- This paper presents comprehensive experiments on comparing the proposed method with existing positional encoding approaches.
- The paper is well written and easy to understand.

Cons:
- It would be helpful to discuss the potential applications of the proposed method other than language modeling.
- I am curious about comparing the transformer+ALiBi with LSTM in extrapolation tasks. The idea of adding temporal bias to attention is similar to the forget gate in LSTMs. Therefore, adding LSTM as a reference will make the paper stronger.

**Update:** The additional results are convincing. I raised my rating to acceptance.

**Summary Of The Paper:**

This paper investigates the extrapolation capability of transformer-based language models. The authors observed that existing positional encoding methods (e.g., sinusoidal embedding, relative positional embedding) fail to generalize to longer sequences in language modeling tasks. Therefore, they introduce a new positional encoding method called ALiBi, which adds temporal bias to the multi-head attention to penalize attention score proportional to token distances. Experimental results show that ALiBi has significantly stronger extrapolation capability compared to other positional encoding methods.

**Summary Of The Review:**

This paper proposes an interesting and novel idea for enhancing the extrapolation capability of transformer-based language models. A few additional experiments and discussions will make the paper stronger.

---

> ### Author Response · Authors · 2021-11-18
> **High level vision for ALiBi and comparison to LSTMs**
>
> Thank you for your time and thorough review! We appreciate that you found ALiBi to be interesting and novel and the experiments to be comprehensive!
> Response to the cons:
> 1. We agree with your statement and will update the next version to make sure that our overall high-level vision is clear. We believe that language models are the current backbone of the NLP world, and therefore finding ways to make them better benefits the whole field. For example, GPT-3 is an incredibly strong model for multiple different tasks, but training it is prohibitively expensive. We believe that methods like ALiBi that cut down the training time of large LMs could one day lead to more affordable training of GPT-3-sized models, which would have an immense benefit to the NLP community. In addition, ALiBi provides the benefit of being able to read and also generate sequences that are longer than the ones seen during training. Lastly, in our response to reviewer dz7S we included some new results on downstream tasks, showing that ALiBi also helps there.
> 2. We agree that LSTMs are a natural model to compare to, since as you mention the forget gate biases them to be dependent on the recent past and to forget the distant past, similar to ALiBi. As we note in the introduction, in the pre-Transformer era, when language modeling was primarily using LSTMs, extrapolation was something we all got for free right out of the box.
> Looking at the WikiText-103 leaderboard at PapersWithCode [1], the best RNN-based models are the LSTM from “Fast Parametric Learning with Activation Memorization”, the LSTM from “Relational recurrent neural networks” and the QRNN model introduced in “An Analysis of Neural Language Modeling at Multiple Scales”. These models obtain between 29.2 to 36.4 perplexity on the test set, which is much worse than the 18.30 perplexity that our best model obtains on the same corpus. The QRNN model was trained on sequences of length 140 and the other two models were trained on sequences of length 100. So clearly the recurrent models can extrapolate to the much larger test sequence length (since recurrent models always input the previous hidden state, their evaluation subsequence length, L in our paper, is equal to the length of the entire test set). But the performance of these models is just so much worse than contemporary transformers, and so even though these models can extrapolate well, it doesn’t matter since their overall performance is just very weak.
> ALiBi combines the generalization properties of LSTMs with the modelling power of transformers.
> We are grateful that you highlight the extrapolation abilities of LSTMs as we do believe that this is an important point, and we will mention this discussion in the next version of the paper.
>
>
> [1] https://paperswithcode.com/sota/language-modelling-on-wikitext-103

---

### Official Review · Reviewer_ZAja · 2021-10-30

**Correctness:** 3
**Technical Novelty And Significance:** 2
**Empirical Novelty And Significance:** 3
**Recommendation:** 6
**Confidence:** 4

**Main Review:**

At a very high level, I did enjoy the paper as the method is simple and it indeed helps a pretrained transformer-based models to extrapolate to much longer sequences. Some of my concerns were addressed in the authors' response, and the others do require extensive exploration. Therefore, I would like to see this submittion at ICLR2022.


====end of the update====

1. When the dimension of a transformer module is roughly the same as or significantly larger than the number of tokens, the dimension becomes the main contributing factor to the time complexity, which explains why, with the linear bias, the model only achieved ~10% speedup.

2. I was wondering if we could directly manipulate the probability after the softmax layer, it probably would achieve a similar effect. For example, one can multiply the probability map with a matrix with 1s in the diagonal terms and with linearly decaying off-diagonal terms towards 0, which also effectively biased the model to learn from nearby tokens.

My point here is that the submission could have been more generalised in a way that, say, as long as the bias terms are fixed before training and they have an impact on the attention scores or the probability maps, the model will extrapolate to very long sequences. This would've been a stronger and more generalised message.

3. The title and the intro gave me the impression that it was designed for transformers, but I was wondering whether it would hinder transformers' capability in modelling images or biological sequences where tokens that are far from the current one would still play an important role.

For images, the current approach of serialising an image is either at pixel-level or at patch-level, which means that tokens surrounding the current one in 2-dimensional space will be the context, however, the proposed approach would potentially worsen the situation.


**Summary Of The Paper:**

The submission proposed an effective approach to allow pre-trained transformer-based language models to extrapolate beyond the maximum length used in training, which potentially reduces the training time as extrapolation is empirically guaranteed. The proposed method adds fixed biases to the dot-product values between queries and keys that linearly decays w.r.t. the gap between two positions. Empirically, the proposed method indeed successfully allows pre-trained models to be evaluated on sequences that are multiple times longer than the training ones without significant loss.

**Summary Of The Review:**

The submission proposed a simple yet effective method that helps pre-trained language models to extrapolate beyond the sequence length used in the training, but I think the paper could've delivered a stronger message. I am open to discussions.

---

> ### Author Response · Authors · 2021-11-18
> **ALiBi's benefits and applications to other domains**
>
> Thank you for your time and thorough review! We are grateful that you found ALiBi to be simple yet effective!
> 1. You correctly state that on very large models, ALiBi leads to around a ten percent training speedup. With GPT-3 estimated to have cost many millions of dollars to train, we do believe that a ten percent speedup is a significant benefit. Although we think this contribution (ALiBi’s extrapolation abilities allowing for training on shorter sequences thus saving training time) is the most interesting one in our work, we believe that we have two other strong contributions that should excite ICLR readers even if they don’t care about a 10% speedup. The first of these is that we show for the first time that existing language models cannot extrapolate! A few previous papers (including Vaswani et. al. 2017) had mentioned that transformers theoretically had the ability to extrapolate, but we show for the first time that they don’t do it. Our second contribution is that we present a method (ALiBi) that replaces the complex and unintuitive sinusoidal position embeddings with an extremely simple and efficient alternative, and performs either on-par or even better than the sinusoidal embeddings. So even if a reader does not care about efficiency gains, we strongly believe that they would be excited by this drop-in position method replacement that is both intuitive and works great in practice.
> 2. That’s a great question! In fact, we did try this approach of multiplying instead of adding the bias (with the bias being in the range of 1 to 0), and we found it didn’t work. We mentioned this at the end of the first paragraph in the related works section. We agree with you that this is an intuitive idea that seems like it should work, and we were surprised when it didn’t.
> 3. You correctly state that ALiBi biases attention values to decrease as the distance between the tokens increases, but it’s important to note that ALiBi does not totally disable attention between tokens that are far away. We demonstrate improvements in language modeling from ALiBi even when including thousands of tokens of context.
> You brought up the vision and biological sequence domains, which are two domains which we are excited about and believe ALiBi could be applied to.
> You are correct that the current proposal gives a better inductive bias for text than images, however there are natural 2d generalizations that we will explore in future work.

---

> > ### Comment · Reviewer_ZAja · 2021-11-22
> > **thanks for the answers!**
> >
> > I am overall satisfied with the answers and will raise the rating accordingly.

---

> ### Author Response · Authors · 2021-11-22
> **Last day of author response**
>
> Dear reviewer ZAja,
>
> As today is the last day for author response, we would be very grateful if you could specify if we've answered your questions and concerns.
>
> Thank you!

---

### Official Review · Reviewer_6wKQ · 2021-11-02

**Correctness:** 4
**Technical Novelty And Significance:** 3
**Empirical Novelty And Significance:** 3
**Recommendation:** 8
**Confidence:** 5

**Main Review:**

The method is simple and quite effective. The paper addresses an important research problem of input length extrapolation. ALiBi developed on Wikitext-103 generalizes to 1.3B parameter model. ALiBi’s inductive bias also improves the accuracy.

Previous works did not rigorously evaluate the extrapolation of a transformer and simply assumed the possibility of extrapolation. On the other hand, this paper carefully measured extrapolation compared with other works (Rotary and T5 Bias) and devised their own method to overcome the limitations of previous works. The method itself might look less novel or incremental because previous works inspire its many parts. It would be much better to provide theoretical explanations more than empirical proof on why ALiBi enables better extrapolation and higher final accuracy.

ALiBi is only evaluated on language modeling in this paper. A transformer is a widely used neural architecture for many different tasks and domains. They also mentioned in the related work section that other works studied extrapolation on other tasks. I wonder about the authors’ thoughts whether their ALiBi could be helpful to other tasks as well. Of course, the importance of the longer context and extrapolation ability may vary depending on the task.

One minor question is that the dot products of queries and keys are usually divided by the square root of the dimension, and it is maybe abbreviated in the equation. I am curious this division is performed after or before adding a bias.

Each head has a different slope for the linear bias, so I expect that heads learn different patterns. An analysis of that would be interesting. The authors argue that the method is robust to slope choice, but they found that other alternatives underperform, such as learning these slopes. Because many other design choices are possible, I am curious how they found the final solution and what they tested.

ALiBi was tested on two different model sizes. According to their results, extrapolation on a billion language model (improving until ~2x) is relatively inferior to that on a Wikitext-103 scale language model (improving until ~6x). I worry whether extrapolation ability reduces as the model becomes bigger (or with more training data).

**Summary Of The Paper:**

The paper addresses the extrapolation problem where a test sequence longer than training sequences is given and proposes Attention with Linear Biases (ALiBi) that adds a penalty linear to the distance between a query and a key to the attention scores. ALiBi shows remarkable input length extrapolation ability while computationally efficient with almost marginal overhead compared to the standard transformer. Moreover, ALiBi does not induce any additional parameters and generalizes well to a billion scale language model.

**Summary Of The Review:**

The paper is well written and easy to follow. The contribution is concrete and practically useful since a transformer is a building block of many machine learning models. More importantly, the size of language models becomes bigger, so their training cost is prohibitive. ALiBi improves the efficiency of language model (or transformers in general) training.

---

> ### Author Response · Authors · 2021-11-18
> **Response to 6wKQ**
>
> Thank you for your time and thorough review! We’re happy you found the research problem we tackled to be important and that you found our solution to be useful and efficient!
> 1. We agree that understanding why and how ALiBi works is important, and the analysis section in our appendix provides an initial exploration, where we show that ALiBi improves performance by reducing the early token curse. Of course, there is lots more work to be done here and we hope that this inspires others in the community to explore ALiBi and other, similar, methods as well.
> 2. Language modeling was just our initial proving grounds, as we believe that LMing is a challenging task and since it is useful for so many downstream tasks. But our aim with ALiBi is to find a method that can work for multiple tasks. We’ve already received reports from two different groups that have managed to make ALiBi work in machine translation, where it leads to models that are able to extrapolate to longer sequences. We’ve recently obtained interesting preliminary results in speech recognition, and we are planning on exploring ALiBi on a wide set of diverse transformer models in the future.  We believe these are out of scope for the current paper.
> 3. In our response to reviewer dz7S we included some new results on downstream (language modeling) tasks, showing that ALiBi also helps there.
> 4. Your question about the sqrt(dim) factor is indeed one that we should have more clearly answered in the paper- the ALiBi bias is added *without* being multiplied by the sqrt(dim) factor. We will definitely clarify this in the next version of our paper.
> 5. Since the paper was released we’ve conducted some preliminary analysis showing that indeed the different slopes cause the heads to learn significantly different attention patterns and we might add this to the appendix of the next version of this paper.
> 6. To clarify- we indeed found other static (non-learned) slope patterns to work well, but we could not quite get learned slopes to extrapolate well. We found the set of slopes that we presented in the paper by initially training the slopes. That model with trained slopes got good perplexity when not extrapolating but could not extrapolate at all. The slopes that we obtained from that experiment were all between 0 and 1 with their frequency increasing the distance to 0 decreased (similar to the slopes obtained by our equation). Just that a single slope did not follow these constraints- it was a negative number. Changing it to a positive number between 0 and 1, and retraining the model, enabled extrapolation. Then we just tried a few different methods that fit this constraint that we discovered (all slopes should be between 0 and 1, with the frequency increasing as you get closer to 0). We tried generating slopes from the exp(0.5) or pareto(1) distributions, and these worked, but had high variance. So then we settled on the method shown in the paper of starting at .5 and multiplying by .5 to obtain the next slope. (That was for models with 8 heads). Then, when we needed 16 heads we just geometrically interpolated the 8 slopes that we had so that we got 16 slopes.
> 7. As the reviewer correctly states, and as we noted in the paper, our results on the 1.3B parameter models trained on the big corpus are very strong but not as strong as our results on the smaller models trained on the much smaller WikiText-103 corpus. The inductive bias imbued by ALiBi seems extremely useful for smaller-scale models, but as with all inductive biases, it becomes less impactful as we increase model size, training time and training data. Our results on the large models are still very impressive, and lead to significant efficiency gains, and we hope that our paper will ignite a conversation about extrapolation that will lead to even greater gains for big models in the future.

---

> > ### Comment · Reviewer_6wKQ · 2021-11-30
> > **Thanks for the thorough response**
> >
> > Most of the answers addressed my concerns and unclear parts.

---

### Official Review · Reviewer_dz7S · 2021-11-02

**Correctness:** 4
**Technical Novelty And Significance:** 3
**Empirical Novelty And Significance:** 3
**Recommendation:** 8
**Confidence:** 4

**Main Review:**

Strengths:
- To my knowledge, the paper is the first to study length extrapolation in Transformer language models. This is an important open problem for language modeling.
- The proposed ALiBi mechanism is simple to implement and computationally efficient.
- Experiments confirm that the proposed method enables length extrapolation for language modeling.
- The paper is well-written and easy to follow.

Weaknesses:
- Experiments can be expanded. I am curious if the findings also apply to other tasks, such as text classification, sequence labeling, and sequence-to-sequence generation. The proposed method is simple to implement, so I imagine it would not be hard to add a few more tasks.

Missing related work:
- Xu et al., 2021. How neural networks extrapolate: from feedforward to graph neural networks. This paper studies a similar kind of input length/size extrapolation for graph neural networks.

**Summary Of The Paper:**

This paper studies input length extrapolation for Transformer language models; i.e., how Transformer LMs perform on test sequences that are longer than training sequences. The paper finds that how positions are encoded plays a crucial role for input length extrapolation. Models with sinusoidal and rotary position embeddings do not extrapolate well, while T5’s position-dependent attention mechanism (dubbed T5 bias) enables better extrapolation. The paper then proposes ALiBi, another attention mechanism that also allows extrapolation while being computationally more efficient than T5 bias. These results are empirically confirmed on two datasets.

**Summary Of The Review:**

The paper studies a novel problem, input length extrapolation in language modeling, and proposes a simple solution with good empirical results. The paper is also well-written. One way to further improve the paper is to add experiments on other tasks. Overall, I recommend acceptance of this paper.

---

> ### Author Response · Authors · 2021-11-18
> **New zero-shot results for 2.7B param models**
>
> Thank you for your time and thorough review! We sincerely appreciate your remarks on the importance of our research problem and the simplicity and efficiency of our solution!
> 1. We agree that experiments on downstream tasks and other transformer tasks (such as sequence-to-sequence generation) are the correct next step, and we’ve started working on these already. We have some new results on zero-shot evaluation that we can share with you. We trained a 2.7B parameter baseline LM, trying to match the settings of the 2.7B GPT-3 model as closely as possible (32 layers, 32 heads, 2560 emb dim, 2048 input tokens). We then trained a new model, using the same hyperparams, except that we use ALiBi and train on sequences of length 1024 (so training is 20% faster).
> We evaluated both models on seven zero-shot tasks, following the same evaluation methodology as used in GPT-3 (page 63 of the GPT-3 paper shows the results they obtained on these tasks).
> |                 | Baseline (2.7B params trained on 2048 tokens) | ALiBi (2.7B params trained on 1024 tokens) | Diff  |
> |-----------------|-----------------------------------------------|--------------------------------------------|-------|
> | storycloze      |                                     __78.25__ |                                      78.19 | -0.05 |
> | openbookqa      |                                         49.60 |                                  __52.20__ |  2.60 |
> | piqa            |                                     __76.55__ |                                  __76.55__ |  0.00 |
> | winogrande      |                                         61.40 |                                  __63.54__ |  2.13 |
> | arc (easy)      |                                     __60.88__ |                                      58.25 | -2.63 |
> | arc (challenge) |                                         28.43 |                                  __29.10__ |  0.67 |
> | hellaswag       |                                     __65.87__ |                                      65.46 | -0.42 |
> Our results show that the ALiBi model is able to overall slightly outperform the baseline even though it is faster to train, showing that ALiBi is useful on multiple downstream tasks and not only when evaluating perplexity.
>
>
> 2. Thank you for the related work pointer on extrapolation! We will take a close look and draw connections where appropriate.

---

> > ### Comment · Reviewer_dz7S · 2021-11-19
> > **Thanks for new results**
> >
> > The new results look very good and can definitely improve the paper. Thanks for adding these!
> >
> > I have no further questions at this point. It would be even better to have some curves for these tasks (similar to Figure 6), but I understand this takes time and compute, so I am happy with the current results.

---

### Author Response · Authors · 2021-11-19
**Discussion period ends soon**

Dear reviewers-

As the discussion period ends soon, we would be grateful to know if our responses answered your questions and concerns.

Our responses showed that ALiBi efficiently provides benefits for downstream tasks at the 2.7B parameter scale, we compared ALiBi to LSTMs as requested by reviewer xt6W, and reiterated the advantages of ALiBi beyond just extrapolation.

Thank you for your time and effort!

---

### Decision · Program_Chairs · 2022-01-20

**Decision:**

Accept (Poster)

**Comment:**

This submission proposes a simple, efficient, and effective position representation method for the Transformer architecture called ALiBi. ALiBi enables better extrapolation and performance (in terms of efficiency and task performance). The submission also includes careful analysis and extensive experiments, and notably suggests that the gains of ALiBi may be less pronounced in more scaled-up settings. All reviewers agreed the paper should be accepted. I think it's reasonably likely that ALiBi will become a common choice in future Transformer models, or at the very least that this work will prompt further work on developing improved position representations for Transformer models. I therefore recommend acceptance.